# Non-Tuberculous Mycobacteria and *Aspergillus* Lung Co-Infection: Systematic Review

**DOI:** 10.3390/jcm11195619

**Published:** 2022-09-23

**Authors:** Marina Fayos, Jose Tiago Silva, Francisco López-Medrano, José María Aguado

**Affiliations:** 1Unit of Infectious Diseases, Department of Internal Medicine, University Hospital 12 de Octubre, 28041 Madrid, Spain; 2Instituto de Investigación Hospital 12 de Octubre (Imas12), 28041 Madrid, Spain; 3Centro de Investigación Biomédica en Red CIBERINFEC, ISCIII, 28029 Madrid, Spain; 4Department of Medicine, School of Medicine, Universidad Complutense, 28040 Madrid, Spain

**Keywords:** non-tuberculous mycobacteria, *Aspergillus*, chronic lung disease

## Abstract

Non-tuberculous mycobacteria (NTM) and *Aspergillus* pulmonary co-infection occurs in patients with underlying lung disease and is rarely reported. We conducted a systematic search of NTM and *Aspergillus* pulmonary co-infection in PubMed, EMBASE, and Cochrane Library to identify cases published from 1977 to May 2022. We included 507 articles comprising 1538 cases (only 817 patients with partial relevant clinical data). Of these, 54.3% of patients were men, with a mean age of 57.7 years. Chronic obstructive pulmonary disease (21.1%), previous diagnosis of tuberculosis (18%), and asthma (11.1%) were the most common chronic lung diseases, and corticosteroids were used in 36.8% of patients. The most frequent symptoms were cough (68.2%), dyspnea (59.1%), and hemoptysis (34.1%). The most common radiological findings were bronchiectasis (52.3%) and cavitation (40.8%). NTM and *Aspergillus* were treated simultaneously in 47.3% of cases, whereas NTM-targeted therapy only was performed in 23.4% and *Aspergillus* only in 1.6%. The remaining 27.7% did not receive any treatment and were considered to be colonized. The global mortality rate was 43% (159/370). There was an increased prevalence of NTM and pulmonary aspergillosis among patients with underlying chronic lung diseases, which led to severe pulmonary affection with a poor global prognosis.

## 1. Introduction

*Aspergillus* causes different clinical manifestations, from colonization to pulmonary aspergillosis, according to the immune competence of the patient [1]. Chronic pulmonary aspergillosis (CPA) is a progressing lung disease caused by *Aspergillus* species, which affects patients with underlying lung disease [2]. CPA includes different disease manifestations, including simple aspergillomas, *Aspergillus* nodules, chronic cavitary pulmonary aspergillosis (CCPA), and chronic fibrosing pulmonary aspergillosis. CCPA is characterized by one or more pulmonary cavities with or without aspergillomas [3].

Non-tuberculous mycobacterial (NTM) pulmonary infection is increasing worldwide [4,5]. Mycobacterial infections may also cause diverse pulmonary manifestations, and patients with NTM often have co-infections with numerous pathogenic microorganisms, including *Aspergillus* [6]. Both *Aspergillus* and mycobacterium are opportunistic pathogens that can cause severe lung disease, but co-infection is reportedly associated with poor prognosis [7,8], and it is rarely reported in the literature. The delayed diagnosis of this condition and the challenge in management due to drug interaction may lead to a worse prognosis [2].

We have reviewed the cases of co-infection by NTM and *Aspergillus* described in the literature to establish the predisposing factors for developing the disease, the most common clinical manifestations, and the consequences of the evolution of the disease in these patients. This knowledge may be useful for providing an earlier diagnosis and better management of these patients.

## 2. Materials and Methods

### 2.1. Search Strategy

A systematic search was performed using PubMed (Medline), EMBASE, and Cochrane Library databases from inception to 24 May 2022. The following terms were used to identify related publications: “mycobacterium infections, non-tuberculous” or “non-tuberculous mycobacteria” or “atypical mycobacteria” and “*aspergillus*” or “aspergillosis”. We included literature in English, French, or Spanish (Figure 1).

We included published articles that included the following criteria: case reports, case series, and case-control or cohort designs. We included those patients in whom NTM and *Aspergillus* had been concomitantly isolated from a respiratory sample, and both clinical and microbiological criteria were compatible with NTM and aspergillosis.

In order to identify the NTM cases, clinical, radiologic, and microbiologic criteria for diagnosis of Non-tuberculous Mycobacterial Pulmonary Disease were used, based on the last Infectious Diseases Society of America (IDSA) guidelines [9]. We included patients with pulmonary or systemic symptoms and the following radiologic findings: nodular or cavitary opacities on a chest radiograph or bronchiectasis with multiple small nodules in a high-resolution computed tomography scan (CT) [9]. All subjects also met the following microbiologic criteria: positive culture results from at least two separate expectorated sputum samples or one positive culture from at least one bronchial wash or lavage or lung biopsy with mycobacterial histologic features (granulomatous inflammation or AFB) and positive culture for NTM or biopsy showing mycobacterial histologic features (granulomatous inflammation or AFB) and one or more sputum or bronchial washings that are culture positive for NTM [9].

Pulmonary aspergillosis cases were identified and categorized as invasive, chronic, or allergic according to the last Treatment of Aspergillosis Guidelines from IDSA [10]. Invasive disease caused by *Aspergillus* species includes infection of the lower respiratory tract as the portal of entry and other tissues that may be infected as a result of hematogenous dissemination or direct extension from contiguous foci of infection [10]. Chronic forms of pulmonary aspergillosis consist of simple aspergilloma, CCPA, and chronic, necrotizing pulmonary aspergillosis (CNPA) [10]. CCPA is defined as multiple cavities with or without aspergilloma in association with pulmonary and systemic symptoms and an increase in inflammatory markers [10]. Allergic conditions, such as allergic bronchopulmonary aspergillosis, were also included [10].

Studies were excluded if they did not fulfill these criteria and included only NTM or aspergillosis cases in the absence of co-infection, extrapulmonary diseases such as skin or soft-tissue infection, or cases that were diagnosed in animals. In the case of partially overlapping publications, only the study with the highest number of individuals or the most modern series was included.

We reviewed and extracted data using a standard process for each retrieved article. The following variables were assessed using a standardized data collection form: demographics, alcohol and smoking habits, comorbidities, clinical presentation, type of pattern on chest imaging studies (simple radiography or CT), microbiologic findings, type of infection, treatment, and outcome.

### 2.2. Statistical Analysis

Only descriptive analyses are presented in this review. Quantitative data are shown as the mean standard deviation (SD) or the median with absolute or interquartile (Q1–Q3) ranges. Qualitative variables are summarized with absolute and relative frequencies and 95% confidence intervals (CIs). The descriptive analysis was carried out using SPSS v. 15.0 (SPSS Inc., Chicago, IL, USA).

## 3. Results

### 3.1. Results of the Search

Overall, we retrieved 763 articles. First, we excluded papers in languages other than English, French, or Spanish (12 articles). After a detailed evaluation, we also excluded those papers that did not refer to NTM and *Aspergillus* co-infection (67 articles), extrapulmonary disease cases (10 articles), or those specifically involved with treatments (5 articles). We also excluded 45 literature reviews with no original data, 5 studies and reports based on animals, and 2 guidelines. A total of 50 references were not included due to the impossibility of obtaining the complete article. Therefore, we included a total of 507 articles comprising 1538 cases of NTM and *Aspergillus* co-infection published since 1977. Of these cases, 40 were individually detailed cases, and 1498 were summarized in reports containing 2–187 cases. There were 817 patients for whom there was only partial relevant clinical or microbiological information, and they were included in this study.

### 3.2. Demographics and Comorbidities

We analyzed a total of 817 cases with enough relevant, although partial, clinical or microbiologic data. Basic demographics (age and sex) were available for 523 and 521 of these patients, respectively; 54.3% were men, with a mean age at diagnosis of 57.7 +/− 20 years. Major lung and other chronic comorbidities are detailed in Table 1. Chronic obstructive pulmonary disease (COPD) was the most frequent chronic lung disease (21.1%, *n* = 105), followed by a previous diagnosis of active tuberculosis (18%, *n* = 92), asthma (11.1%, *n* = 55), interstitial lung disease (6%, *n* = 30), and emphysema (4.8%, *n* = 24). Corticosteroids were used in 36.8% of cases (*n* = 183); 18.7% inhaled (*n* = 93) and 18.1% systemic (*n* = 90).

### 3.3. Clinical Presentation

Clinical presentation was only available for 44 patients and is described in Table 2. The most common symptom was cough (68.2%, *n* = 30), followed by dyspnea (59.1%, *n* = 26), hemoptysis (34.1%, *n* = 15), and fever (22.7%, *n* = 10). Chest imaging studies (simple radiography or CT scan) were described in 512 cases. The most frequent findings were bronchiectasis (52.3%, *n* = 269), cavitation (40.8%, *n* = 209), and nodules (36.7%, *n* = 188). These lesions were more frequently found in the middle lobe (31%, *n* = 18) and in the right upper lobe (29.3%, *n* = 23).

### 3.4. Diagnostic Approaches

All cases in this review were cataloged as NTM and *Aspergillus* co-infection, but culture results for microbiologic diagnosis were reported only in 585 patients for NTM and in 463 patients for *Aspergillus*. The microbiological species most frequently isolated from NTM and *Aspergillus* are detailed in Table 3. Microbiologic-based diagnosis was made earlier for NTM than for *Aspergillus* in 82.8% of cases (*n* = 67). The most common NTM species was *Mycobacterium avium* complex (MAC [73.2%, *n* = 428]), followed by *Mycobacterium abscessus* complex (10.4%, *n* = 61). The most common *Aspergillus* species was *A. fumigatus* (65.7%, *n* = 304) followed by A. *niger* (12.3%, *n* = 57). *Aspergillus*-specific IgG was reported and positive in 15.9% of all cases (*n* = 132) and *Aspergillus*-specific IgE in 16.7% (*n* = 138;). Galactomannan and Beta-D-Glucan were rarely reported and were positive in 1% (*n* = 9) and 0.8% (*n* = 7) of all cases, respectively. Negative microbiologic results were also rarely reported, so the positive rate of all tests performed could not be properly calculated.

The following forms of different Aspergillosis infections were reviewed. CPA was the most common manifestation, reported in 42.6% (*n* = 86), followed by Allergic Bronchopulmonary Aspergillosis (ABPA) in 17.8% (*n* = 36), CNPA in 10.9% (*n* = 22), and invasive aspergillosis (IA) in 1% of cases (*n* = 2). *Aspergillus* colonization was presumed and reported in 51 cases (6.2%).

### 3.5. Treatment and Outcome

Data on anti-mycobacterial therapy were available for 189 cases and for antifungal therapy in 243 cases. The drug regimens in the patients that received anti-tuberculosis and antifungal therapy are detailed in Table 4. NTM and *Aspergillus* infection was treated simultaneously in 47.3% of cases (*n* = 172), whereas only NTM-targeted therapy was performed in 23.4% (*n* = 85) and only for *Aspergillus* in 1.6% (*n* = 6). There was no medical therapy for NTM or *Aspergillus* in 27.7% of cases (*n* = 101), assuming both were colonization. The most common anti-tuberculosis regimen consisted of rifampicin, ethambutol, and a macrolide, and the most frequently used antifungal drug was itraconazole (31.7%, *n* = 77). Systemic corticosteroids were used in 9 cases (3.7%). A surgical approach, such as lobectomy or nodulectomy, was performed in 14 patients (5.8%), and aneurysm embolization was necessary in 19 cases (7.8%). Data on hospital discharge and outcome were reported in 370 patients (44.8%), and the mortality rate was 43% of these patients (159/370) after a median of 2.7 months from the diagnosis (IQR 1-27.5). The outcome was reported as satisfactory in 203 patients (54.9%).

## 4. Discussions

Pulmonary NTM and Aspergillosis lung disease are both infectious diseases associated with a high mortality rate. The incidence of co-infection of pulmonary aspergillosis (PA) and NTM lung diseases may be underestimated, especially due to uncertainty of the diagnosis of NTM lung diseases. A relationship between NTM and *Aspergillus* has been increasingly reported, but the explanation for this occurrence is not well understood.

Several hypotheses have been raised to explain this association. Firstly, NTM-infected patients with underlying structural lung diseases have an increased risk of fungal colonization, and this may ultimately cause *Aspergillus* lung infection. NTM lung disease is also frequently detected in immunocompromised patients [11] and in patients who were diagnosed with tuberculosis (TB) and received treatment for it [12,13]. TB also increases the risk of *Aspergillus* infection. Furthermore, NTM lung infection generates destructive lung lesions such as lung cavitation, which increments the incidence of PA [12,13].

In this review, we found that the conditions most frequently associated with PA and pulmonary NTM disease are fibrocavitary lesions, bronchiectasis, the presence of COPD with pulmonary emphysema or asthma, steroid use, and previous active tuberculosis. These findings are similar to what has been described in the literature for the last few years. [8,14,15]. PA is more frequent in patients who have previous TB cavitation. This can be interpreted as *Aspergillus* colonization and is more common in patients with previous chronic lung cavities. The time between primary TB infection and PA was not specified in this review since the data were poorly reported. However, it may be interesting to analyze the time relation between both events, to enhance the ability to diagnose and treat these patients. Long-term systemic and inhaled corticosteroid treatment also disposes toward this co-infection. Corticosteroids debilitate the innate and adaptive host immune systems [16], which aggravate the ability to eliminate the fungal infection, and lead to the growth of *Aspergillus* hyphae [16,17,18].

A diagnosis of PA and NTM pulmonary infection requires the presence of clinical characteristics, consistent radiological findings, isolation of NTM and *Aspergillus* infection, and exclusion of other possible differential diagnoses. The most frequent symptoms found in our review were cough, dyspnea, hemoptysis, fever, and weight loss, similar to the reported literature [19]. The clinical characteristics of these two diseases are very similar, therefore, microbiological and radiological findings are essential to make the diagnosis. PA and NTM co-infection radiological findings include the thickening of pre-existing lung cavities surrounded by infiltration and the presence of a fungal ball or aspergilloma [20]. In our review, the most frequent locations for these kinds of lesions were the middle lobe and the right upper lobe. The culture positivity of *Aspergillus* in patients with NTM lung infection does not necessarily mean PA, but we found that colonization was assumed in very few cases in this review.

Regarding treatment for PA, we found that the preferred antifungal therapy was triazoles. Treatment of CPA and NTM lung co-infection may be challenging because of the drug interaction between oral triazoles and anti-mycobacterial agents. Most patients require rifampicin as anti-mycobacterial therapy, which is a strong CYP3A4 inducer that may lower the level of the antifungal and, therefore, result in treatment failure [19].

The poor prognosis of NTM and PA co-infection may be explained by PA having been demonstrated as a predictor of mortality in patients with NTM lung disease [21,22]. The high mortality rate in our review was striking. The global mortality was directly related to the NTM and *Aspergillus* co-infection and was more frequently reported in patients with sub-acute aspergillosis, such as CPA. This mortality rate may be related to the immunocompromised status of patients and the use of long-term corticosteroids. Corticosteroid treatment in NTM patients should be used sparingly or avoided, as drug interaction of azoles with rifampicin has been commonly reported, and this may lead to the end of antifungal therapy [23]. Regarding the interactions between azoles and anti-TB therapy, antifungal treatment may be performed first since clinical improvement was more related to the response to antifungal treatment than to the NTM therapy in the cases that were reviewed. Once antifungal therapy is finished, NTM therapy could be started and completed. This way, simultaneous treatment and, therefore, unnecessary pharmacological interactions could be avoided. 

## Figures and Tables

**Figure 1 jcm-11-05619-f001:**
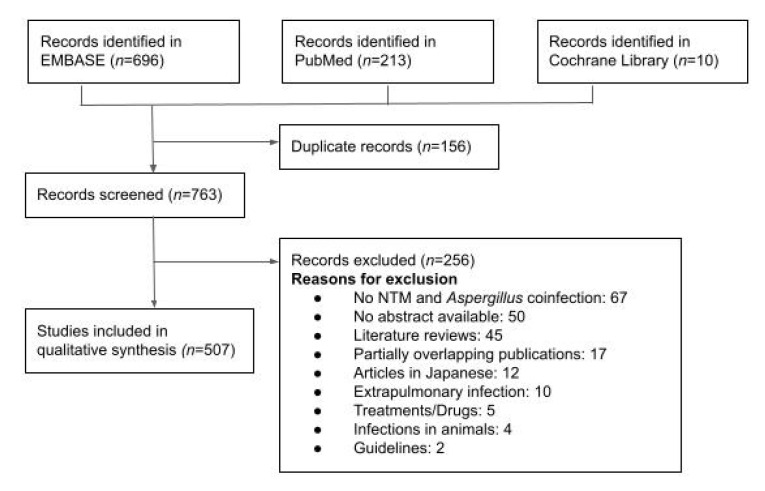
Flow chart for literature search and study selection.

**Table 1 jcm-11-05619-t001:** Demographics and Underlying Conditions in patients with NTM and *Aspergillus* co-infection.

Variable *	No. of Patients (%)
Age, year (mean +/− SD) (521)	54.3 +/− 19
Sex, male (523)	303 (57.7)
Toxic habits (497)	
Active smoking	20 (4)
Ex-smoker	58 (11.7)
Active alcohol use	7 (11.4)
Chronic lung disease (497)	
COPD	105 (21.1)
Previous tuberculosis	92 (18.5)
Asthma	55 (11.1)
Interstitial lung disease	30 (6)
Emphysema	24 (4.8)
Lung transplant	13 (2.6)
Cystic fibrosis	8 (1.6)
Lung cancer	6 (1.2)
Pulmonary fibrosis	5 (1)
Lipid pneumonia	2 (0.4)
COP	2 (0.4)
Sarcoidosis	1 (0.2)
Silicosis	1 (0.2)
Other chronic comorbidities (497)	
Autoimmune disease	19 (3.8)
Rheumatoid arthritis	4 (0.8)
Hematologic cancer	12 (2.4)
Stem-cell transplantation	5 (1)
HIV	6 (1.2)
Corticosteroid treatment (497)	183 (36.8)
Systemic corticosteroids	90 (18.1)
Inhaled corticosteroids	93 (18.7)

* Values in brackets represent number of patients for whom data were available. COPD: Chronic obstructive pulmonary disease. COP: Cryptogenic organizing pneumonia. HIV: human immunodeficiency virus.

**Table 2 jcm-11-05619-t002:** Clinical presentation reported in 44 patients and radiology findings reported in 512 patients with NTM and *Aspergillus* co-infection. Lung lesions locations were mentioned in 58 cases.

Variable *	No. of Patients (%)
Symptoms (44)	
Cough	30 (68.2)
Dyspnea	26 (59.1)
Hemoptysis	15 (34.1)
Fever	10 (22.7)
Weight loss	9 (20.5)
Pleuritic pain	3 (6.8)
Asthenia	2 (4.5)
CT findings (512)	
Bronchiectasis	269 (52.3)
Cavitation	209 (40.8)
Nodules	188 (36.7)
Opacities	39 (7.6)
Aspergilloma	21 (4.1)
Lung lesions location (58)	
Upper lobes	23 (39.7)
Right upper lobe	17 (29.3)
Left upper lobe	7 (12.1)
Middle lobe	18 (31)
Lower lobes	16 (27.6)

* Values in brackets represent number of patients for whom data were available. CT = computed tomography.

**Table 3 jcm-11-05619-t003:** Diagnostic Procedures and Microbiologic Findings.

Variable *	No. of Patients (%)
Positive culture for mycobacteria (585) ^†^	
*M. avium* complex °	428 (73.2)
*M. avium*	382 (65.3)
*M. intracelullare*	46 (7.9)
*M. abscessus*	61 (10.4)
*M. kansaii*	37 (6.3)
*M. xenopi*	16 (2.7)
*M. fortuitum*	10 (1.7)
*M. gordonae*	6 (1)
*M. chelonae*	6 (1)
*M. szulgai*	5 (0.9)
*M. malmonese*	5 (0.9)
*M. simiae*	3 (0.5)
*M. massiliense*	2 (0.3)
Other	7 (1.2)
Positive culture for *Aspergillus* (463) †	
*A. fumigatus*	304 (65.7)
*A. niger*	57 (12.3)
*A. flavus*	27 (5.8)
*A. terreus*	5 (1.1)
*Aspergillus* spp.	50 (1.5)

* Values in brackets represent number of patients for whom data were available. ^†^ Number of patients in whom the microbiologic identification was reported. ° *M. avium* complex cultures include *M. avium* and *M. intracelullare* cultures.

**Table 4 jcm-11-05619-t004:** Targeted therapy for NTM and/or *Aspergillus* in 364 patients. Drugs used in 189 patients who received anti-mycobacterial therapy and in 243 patients who received antifungal therapy.

Drugs *	No. of Patients (%)
Targeted therapy (364) ^†^	
NTM and *Aspergillus* therapy	172 (47.3)
NTM therapy only	85 (23.4)
* Aspergillus* therapy only	6 (1.6)
No NTM or *Aspergillus* therapy	101 (27.7)
Anti-mycobacterial therapy (189)	
Macrolide	131 (69.3)
Quinolone	107 (56.6)
Ethambutol	101 (53.4)
Rifampin	74 (39.2)
Amikacin	20 (10.6)
Clofazimine	6 (3.2)
Rifabutin	5 (2.6)
Cefoxitin	5 (2.6)
Tigecycline	3 (1.6)
Linezolid	2 (1.1)
Imipenem	1 (0.5)
Pyrazinamide	1 (0.5)
Streptomycin	1 (0.5)
Cotrimoxazole	1 (0.5)
Doxycycline	1 (0.5)
Antifungal therapy (243)	
Itraconazole	77 (31.7)
Amphotericin B	47 (19.3)
Voriconazole	24 (9.9)
Embolization	19 (7.8)
Surgery	14 (5.8)
Corticosteroids	9 (3.7)
Caspofungin	3 (1.2)
Posaconazole	2 (0.8)
Isavuconazole	2 (0.8)
Micafungin	2 (0.8)

* Values in brackets represent number of patients for whom data were available. ^†^ Number of patients in whom the targeted therapy for NTM and/or *Aspergillus* was performed.

## Data Availability

Data sharing is not applicable to this article as no new data were created or analyzed in this study.

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
