# Peer review of "Non-Tuberculous Mycobacteria and Aspergillus Lung Co-Infection: Systematic Review"

_jcm, 2022, doi:10.3390/jcm11195619_

Round 1
Reviewer 1 Report
This manuscript is quite good, just minor revision is needed.
1. This manuscript is quite good
2. The arrangement of writing is good
3. The method of systemic review is correct
4. The English is good
5. Need minor correction:
a. Writing the table (on the manuscript)
b. Using words (drug interaction)

Author Response
Thank you for your comments and suggestions.
I am sorry, but I did not fully understand the modifications you suggest about writing the table. I changed some format issues, though.
Please see the attachment.

Reviewer 2 Report
This manuscript presents interesting data on the update of global prevalence of NTM and Aspergillus lung co-infection. However, it could be better and easier to the readers, if the authors provide some additional information. Furthermore, some corrections or clarifications are needed:
1. Please explain the methods to identify the NTM cases clearly, as well as the type of Aspergillus lung co-infection cases
2. The authors need to clarify the diagnostic step (possibly in the form of flow charts) to make it clearer for the readers
3. It is better to put the data based on geographic area to know the global distribution of the cases (if it is possible)
4. Please carefully describe about global mortality rate (line 21) : Global mortality rate was 43% (159/370) ïƒ is it directly related to the NTM & Aspergillus co-infection or other comorbidities ?
Could you explain what kind of population and the disease spectrum of aspergillosis related to the mortality cases (invasive? sub-acute? critically ill patients or others?
5. Please be consistent in writing the species name and some exact terms, such as: Aspergillus (in italic), aspergillosis, etc.
Author Response
Thank you so much for your comments and suggestions.
I made some modifications on the manuscript according to your indications. However, I'm afraid there were no sufficient data to describe a geographic distribution of the cases.
Please see the attachment.

Reviewer 3 Report
Manuscript by Fayos Pérez et al. shows a systematic review to identify cases of Aspergillus fumigatus and NTM respiratory coinfections. The report is clear and concise but some additional information is required to clarify some points.
1) Introduction: The introduction section could be expanded a bit to provide with a framework for the research done in this manuscript. At the end of the introduction the authors say the aim is to review the literature to establish consequences of confection in patient outcome but this is not the same objective stated in the abstract and it is not in agreement with the data presented in the results sections. Authors should unify the message they want to tell with this publication.
2) Methods: The authors used several research databases to perform their literature search. Did the authors adhere to the PRISM guidelines to conduct systematic reviews? If so, authors should cite the relevant publications. In addition, some of the data presented in the methods as literature search should be in an independent section in the results. The results of the search is results section per se no methods. In the data analyses section, the authors mentioned that statistical analyses were carried out using SPSS but there are not statistical analyses done. If only descriptive analyses they should specify this, if some analyses were carried out these should be shown and explained.
3) Results: The results section follows a logical flow, it is easy to read and tables are appropriate. As authors claimed coinfection occur in the setting of prior tb, how long after primary infection does aspergillosis occur? If no enough data maybe this can included as a discussion point. In addition, it is reported that micro-organisms were in some instances isolated. Is there any susceptibility data that can be linked with disease outcome?
Many thanks for submitting this manuscript
Author Response
Thank you so much for your comments and suggestions.
I made some modifications on the manuscript according to your indications. However, I'm afraid there were no sufficient data to describe the susceptibility of microorganisms that you suggest in the results section.
Please see the attachment.
